# One Wave to Explain Them All: A Unifying Perspective on Post-hoc Explainability

## Abstract

Despite the growing use of deep neural networks in safety-critical decision-making, their inherent black-box nature hinders transparency and interpretability. Explainable AI (XAI) methods have thus emerged to understand a model's internal workings, and notably attribution methods also called saliency maps. Conventional attribution methods typically identify the locations – the *where* – of significant regions within an input. However, because they overlook the inherent structure of the input data, these methods often fail to interpret what these regions represent in terms of structural components (e.g., textures in images or transients in sounds). Furthermore, existing methods are usually tailored to a single data modality, limiting their generalizability. In this paper, we propose leveraging the wavelet domain as a robust mathematical foundation for attribution. Our approach, the **W**avelet **A**ttribution **M**ethod (WAM), extends the existing gradient-based feature attributions into the wavelet domain, providing a unified framework for explaining classifiers across images, audio, and 3D shapes. Empirical evaluations demonstrate that WAM matches or surpasses state-of-the-art methods across faithfulness metrics and models in image, audio, and 3D explainability. Finally, we show how our method explains not only the *where* – the important parts of the input – but also the *what* – the relevant patterns in terms of structural components.

## 1 Introduction

Deep neural networks are increasingly being deployed in various applications, such as medicine, transportation, robotics, or finance (Pooch et al., 2020; Sun et al., 2022; Redmon et al., 2016; Thimonier et al., 2024). These networks often make critical decisions, such as detecting tumors in medical images or identifying obstacles in autonomous driving, yet the underlying decision-making process is difficult to interpret due to the black-box nature of the models.

This opacity has motivated the rise of explainable AI (XAI) techniques to provide human-understandable explanations for model decisions. While XAI has been predominantly applied in image classification, it is also extending into other fields, such as audio and 3D shape classification (Parekh, 2023; Paissan et al., 2024; Chen et al., 2021; Zheng et al., 2019).

Among these techniques, feature attribution methods – specifically gradient-based methods for generating saliency maps (heatmaps that highlight important input features, Zeiler & Fergus, 2014) are prevalent. These gradient-based methods (Shrikumar et al., 2017; Sundararajan et al., 2017; Smilkov et al., 2017) considered as efficient and reliable for interpreting model behavior (Crabbé & van der Schaar, 2023; Wang & Wang, 2021; Xue et al., 2023).

Feature attribution involves decomposing a model's decision within a specific "explanation" domain. Traditionally, saliency mapping relied on the pixel domain as this domain. However, pixel-based explanations flatten the hierarchical and spatial relationships inherent in images, effectively collapsing their structural properties. In addition, the pixel domain is only relevant when the input modality is an image. Instead, decomposing the model's decision in the wavelet domain, which preserves the inter-scale dependencies of an input modality, could enable saliency-based methods to account for the image structure in the explanation. Besides, the wavelet domain is defined for any input dimension (images being an input of dimension 2), thus enabling a natural generalization of saliency mapping to modalities such as audio or 3D volumes.

This work introduces the **W**avelet **A**ttribution **M**ethod (WAM), a universal feature attribution method. By unifying and extending existing methods, notably SmoothGrad (Smilkov et al., 2017) and Integrated Gradients (Sundararajan et al., 2017) within the wavelet domain, we enable their application to any modality defined over a continuous space, moving beyond the limitations of the pixel domain. As illustrated in Figure 1, our approach involves computing the gradient of a classification model's prediction with respect to the *wavelet decomposition* of the input signal. We then produce smooth explanations by either averaging over noisy inputs or integrating along the prediction path.

Operating in the wavelet domain, WAM isolates the contribution of the different scales within the input signals to the model's prediction, providing deeper insights into a model's decision-making process. We illustrate these insights by revisiting the meaningful perturbation framework withWAM, or by carrying out noise and overlapping experiments on audio samples. Quantitative evaluations demonstrate that WAM outperforms existing attribution methods across a range of topologies, modalities, and metrics, underscoring its utility in addressing critical challenges in image, audio, and 3D shape classification.

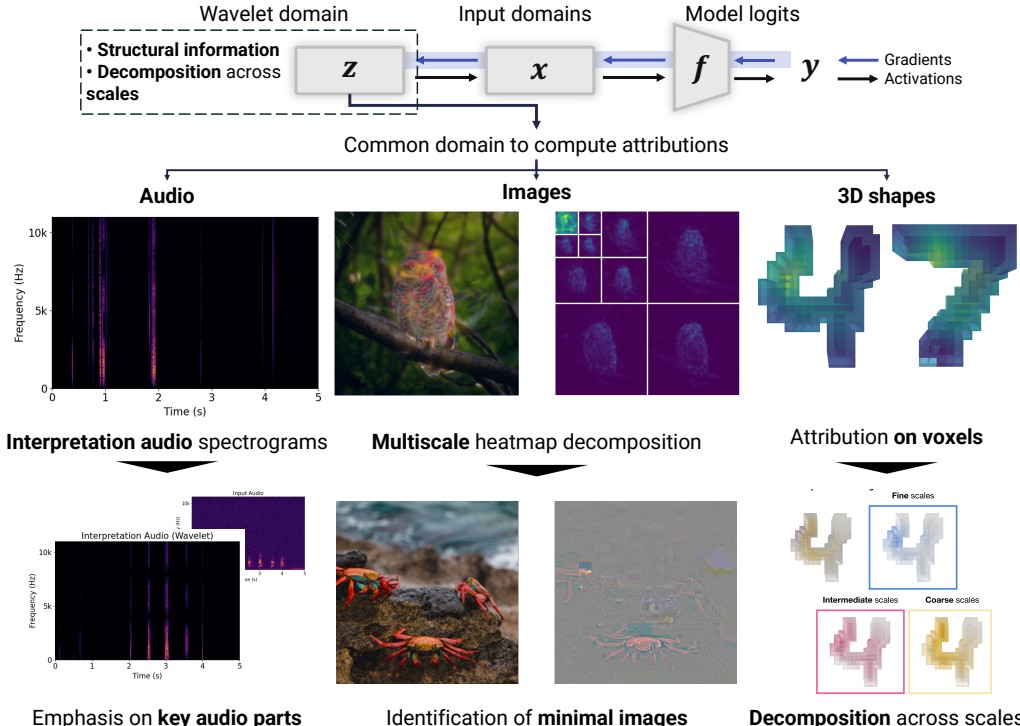

Figure 1: **Explaining any modality by decomposing the model's decision in the wavelet domain.** By computing the gradient of the model's prediction in the wavelet domain, we provide a unified approach to explain the decision of audio, image, and shape classifiers that preserves the structural properties of the input modalities.

## 2 RELATED WORKS

**Images explainability.** Computer vision has supported the development of numerous post-hoc explainability methods (Baehrens et al., 2010), with attribution methods being the most popular. Post-hoc methods are applied on trained model and aim to rank, i.e., estimate an importance for each pixel or region of an image, reflecting its contribution to the score predicted by the model. Many methods have been proposed, which can be classified into two categories: White-box methods, which require access to gradients, and Black-box methods, which use perturbations on the input space.

White-box approaches leverage internal model information, such as gradients, to explain predictions. saliency maps were first introduced by Baehrens et al. (2010) and later refined in Simonyan et al. (2014); Zeiler & Fergus (2014); Springenberg et al. (2014); Sundararajan et al. (2017); Smilkov et al. (2017); or Muzellec et al. (2023). These methods calculate the gradient of the classification score with respect to the input pixels using backpropagation. However, gradients can be noisy in large vision models (Smilkov et al., 2017) and may yield misleading importance estimates due to their focus on infinitesimal input variations (Ghalebikesabi et al., 2021).On the other hand, Black-box methods rely on input perturbations without requiring access to internal model parameters. Techniques like Occlusion (Zeiler & Fergus, 2014), LIME (Ribeiro et al., 2016), RISE (Petsiuk et al., 2018), Sobol (Fel et al., 2021), and HSIC (Novello et al., 2022) generate importance maps by observing changes in the classification score when the input image is altered. For example, Occlusion uses binary masks to systematically occlude regions of the image, while RISE and HSIC apply random masks to perturb multiple regions simultaneously.

However, traditional attribution methods focus on *where* important regions in the image are. However, they fail to address *what* these regions represent in terms of meaningful, higher-level concepts – a gap that more recent research is attempting to fill (Kim et al., 2018; Ghorbani et al., 2019; Fel et al., 2023b; Zhang et al., 2021; Graziani et al., 2023; Fel et al., 2023b;a). Moreover, these methods often under-utilize the inherent structure of images, (recent work are beginning to explore attribution in the frequency domain, e.g. Muzellec et al., 2023).

**Audio explainability.** Previous work on post-hoc audio explainability methods has mainly expanded in three directions. The first explored the use of saliency methods to highlight key features for audio classifiers processing spectrograms (Becker et al., 2024; Won et al., 2019) or 1D waveforms (Muckenhirn et al., 2019). Moreover, while the use of time-frequency representations for classification and explanations is frequent in this regard, wavelet representations have not been explored previously for explanations. The second direction involves variants of LIME (Ribeiro et al., 2016) algorithm, proposed for different types of audio classification tasks (Mishra et al., 2017; 2020; Haunschmid et al., 2020; Chowdhury et al., 2021; Wullenweber et al., 2022). The third has pursued the development of methods to generate listenable interpretations for audio classifiers by leveraging the hidden representations (Parekh et al., 2022; Paissan et al., 2024). LIME-based methods suffer from the issue of high computational costs of explanations due to a large number of forward passes per sample. Recent methods for listenable interpretations require access to hidden layers and train separate modules and are thus unsuitable as post-hoc explainers.

**3D explainability.** 3D data generally comes in two main formats: point clouds and voxels. Point clouds offer an exact representation of the data but are unstructured. Voxels, conversely, are a discretized but structured representation of the data, making them suitable for processing with techniques such as 3D convolutions. Most explainability techniques for 3D data focused on explaining point clouds. Chen et al. (2021); Schinagl et al. (2022); Gupta et al. (2020), and Zheng et al. (2019) introduced techniques to generate visual explanations for interpretability of 3D object detection and classification networks. They highlight critical features in point cloud data by adapting 2D image-based saliency techniques (Gupta et al., 2020; Zheng et al., 2019), by using a perturbation-based approach (Schinagl et al., 2022) or by proposing a 3D variant of LIME (Tan & Kotthaus, 2022). Explainability on 3D volumes remains limited. A few works (Yang et al., 2018; Mamalakis et al., 2023; Gotkowski et al., 2021) have proposed attention maps on 2D slices of 3D medical scans using 3D-GradCAM.

3D and 1D explainability techniques often reproject the model's decision onto a 2D-pixel domain. However, this projection filters out the intrinsic properties of the original signal, such as its temporal or spatial depth, resulting in an incomplete representation. This process, therefore, constitutes an improper way of generalizing attribution methods, as it disregards essential features of the original signal's structure. In addition, the pixel domain itself is limited for explainability. More broadly, we note that the literature has only recently started discussing the expressiveness of the "explanation" domain, e.g., through the lenses of concepts, and still overlooks the broader applicability across modalities. This work contributes to the ongoing discussion by evaluating how the wavelet domain can simultaneously address these concerns.

## 3 METHODS

**Notations & Background.** Throughout, we let $\mathcal{X} = (\Omega, \mathcal{F}, \mu)$ be a measure space with set $\Omega$, $\sigma$-algebra $\mathcal{F}$, and measure $\mu$. We denote by $\mathcal{H} = \mathbb{L}^2(\mathcal{X}, \mu)$ the Hilbert space of square-integrable functions on $\mathcal{X}$. Let $\boldsymbol{f} \in \mathcal{H}$ represent a predictor function (e.g., a classifier), which maps an input $\boldsymbol{x} \in \mathcal{X}$ to an output $\boldsymbol{f}(\boldsymbol{x}) \in \mathcal{Y}$. We denote $\boldsymbol{g} \in \mathcal{H}$ a generic, square-integrable function.

A wavelet is an integrable function $\psi \in \mathcal{H}$ that is normalized, centered at 0, and has zero average (i.e., $\int \psi(\boldsymbol{x}) \, \mathrm{d}\boldsymbol{x} = 0$). Unlike a sine wave, a wavelet is localized in both *space* and *frequency* domains. This localization allows dilations of the wavelet to analyze different frequency intervals (scales) while translations enable analysis at different spatial locations. To compute an image's continuous wavelet transform (CWT), we first define a filter bank $\mathcal{D}$ derived from the original wavelet $\psi$, using a scale factor $\boldsymbol{\lambda} > 0$ and 2D translation $\boldsymbol{b}$. The filter bank $\mathcal{D}$ is given by

$$\mathcal{D} = \left\{ \psi_{\boldsymbol{\lambda}, \boldsymbol{b}}(\boldsymbol{x}) = \frac{1}{\sqrt{\boldsymbol{\lambda}}} \psi\left(\frac{\boldsymbol{x} - \boldsymbol{b}}{\boldsymbol{\lambda}}\right) \right\}_{\boldsymbol{b} \in \mathbb{R}^2, \, \boldsymbol{\lambda} > 0}.$$

The continuous wavelet transform of a function $\boldsymbol{g} \in \mathcal{H}$ at scale $\boldsymbol{\lambda}$ and location $\boldsymbol{x}$ is given by

$$\mathcal{W}(\boldsymbol{g})(\boldsymbol{\lambda}, \boldsymbol{x}) = \int_{-\infty}^{+\infty} \boldsymbol{g}(\boldsymbol{b}) \frac{1}{\sqrt{\boldsymbol{\lambda}}} \psi^*\left(\frac{\boldsymbol{b} - \boldsymbol{x}}{\boldsymbol{\lambda}}\right) \mathrm{d}\boldsymbol{b},$$

which can be rewritten as a convolution (Mallat, 2008). In the discrete dyadic case, the scale factor $\boldsymbol{\lambda}$ takes values in a set $\boldsymbol{\Lambda}$, chosen as $\boldsymbol{\Lambda} = \{2^j : 1 \le j \le N, \, N \in \mathbb{N}, N > 0\}$. Mallat (1989) showed that one can compute the dyadic wavelet transform of a signal $\boldsymbol{g}$ by applying a high-pass filter $H$ to the signal $\boldsymbol{g}$ and subsampling by a factor of two to retrieve the *detail* coefficients, and applying a low-pass filter $G$ and subsampling by a factor of two to retrieve the *approximation* coefficients. Iterating on the approximation coefficients generates a multilevel transform, where the $j^{th}$ level extracts information at resolutions between $2^j$ and $2^{j-1}$ octaves in the frequency spectrum. When the input signal $\boldsymbol{x}$ has dimensionality greater than one, its detail coefficients can be decomposed into different orientations. The common orientations for 2D signals (i.e., images) are vertical, horizontal, and diagonal.

**Wavelets and multiscale decompositions.** Multiscale analysis consists in decomposing an input signal into different levels of detail. The resulting decomposition is particularly interesting as it generates interesting features for signal understanding: edges in images at different orientations and scales correspond to different textures. In sounds, the multiscale decomposition isolates slowly changing patterns from transient ones. Overall, the wavelet decomposition enables the decomposition of an input signal into interpretable components. As we further discuss in section 3.2, the properties of multiscale decompositions translate into several insightful properties for XAI.

### 3.1 GRADIENT-BASED FEATURE ATTRIBUTION IN THE WAVELET DOMAIN

**Problem formalization.** Let $\boldsymbol{f}$ be a classifier and $\boldsymbol{x}$ an input (e.g., an image, an audio, or a 3D shape). The classifier $\boldsymbol{f}$ maps the input to a class $c$ as $\boldsymbol{y}_c = \arg\max_{c \in \mathcal{C}} \boldsymbol{f}(\boldsymbol{x}) \equiv \boldsymbol{f}_c(\boldsymbol{x})$ with a slight abuse of notation. We recall that the original saliency map of the classifier $\boldsymbol{f}$ for class $c$ is then given by $\boldsymbol{\gamma}_{\mathrm{Sa}}(\boldsymbol{x}) = |\nabla_{\boldsymbol{x}} \boldsymbol{f}_c(\boldsymbol{x})|$ where $c$ denotes the class of interest. The saliency map is defined provided that the $\boldsymbol{f}_c$'s are piecewise differentiable (Simonyan et al., 2014). The saliency map highlights the most influential (in terms of the absolute value of the gradient) components in the input $\boldsymbol{x}$ for determining the model's $\boldsymbol{f}$ decision. The higher the value, the greater the importance of the corresponding region.

However, varying pixel values provide no information to what is changing on the image. Therefore, we argue that the pixel domain is not well suited for explaining *what* the model is seeing on the image. On the other hand, the wavelet decomposition of an image – and more broadly of any differentiable modality – provides information on the structural components of the modality. Therefore, computing the gradient of $\boldsymbol{f}$ with respect to the wavelet transform of $\boldsymbol{x}$ will enable us to understand the model's reliance on features such as textures, edges, or shapes in the case of images, transients, or harmonics in sounds or corners or small details in 3D shapes.

Denoting $\boldsymbol{z} = \mathcal{W}(\boldsymbol{x})$ the wavelet transform of $\boldsymbol{x}$, since $\mathcal{W}$ is invertible, we can define the saliency map of in the wavelet domain as

$$\gamma_{\mathrm{Sa}}(\boldsymbol{z}) = \left| \frac{\partial \boldsymbol{f}_c(\boldsymbol{x})}{\partial \boldsymbol{z}} \right| = \left| \frac{\partial \boldsymbol{f}_c(\boldsymbol{x})}{\partial \boldsymbol{x}} \cdot \frac{\partial \mathcal{W}^{-1}(\boldsymbol{z})}{\partial \boldsymbol{z}} \right|, \tag{1}$$

using the fact that $\boldsymbol{x} = \mathcal{W}^{-1}(\boldsymbol{z})$ and where $\dfrac{\partial \boldsymbol{f}_c(x)}{\partial \boldsymbol{x}}$ denotes the gradient of the classifier output with respect to the input image and $\dfrac{\mathcal{W}^{-1}(\boldsymbol{z})}{\partial \boldsymbol{z}}$ is the Jacobian matrix of the inverse wavelet transform. In practice, to retrieve Equation 1, we require the gradients on $\mathcal{W}(\boldsymbol{x})$ and directly evaluate $\partial \boldsymbol{f}_c(\mathcal{W}^{-1}(\boldsymbol{z}))/\partial \boldsymbol{z}$. A remarkable property of this framework is that it **accommodates any input dimension**, and thus it is **modality-agnostic**. Therefore, we can apply it – and leverage its properties – to numerical signals such as audio (1D signals), images (2D signals), or 3D shapes (3D signals). In this paper, we demonstrate the superiority of this method in the 1D and 2D settings compared to other domain-specific methods and illustrate examples for 3D classification.

**Smoothing.** Smilkov et al. (2017) highlighted the fact that the saliency maps computed following Equation 1 can fluctuate sharply at small scales as $\boldsymbol{f}_c$ is not continuously differentiable. To yield smoother explanations, Smilkov et al. (2017) perturb the input image with Gaussian noise. Analogously, we propose to calculate

$$\gamma_{\mathrm{Sg}}(\boldsymbol{z}) = \frac{1}{n} \sum_{i=1}^{n} \nabla_{\tilde{z}} \boldsymbol{f}(\mathcal{W}^{-1}(\tilde{\boldsymbol{z}})) \ \ \text{with} \ \ \tilde{\boldsymbol{z}} = \mathcal{W}(\boldsymbol{x} + \boldsymbol{\delta}) \ \ \text{and} \ \ \boldsymbol{\delta} \sim \mathcal{N}(0, I\sigma^2). \tag{2}$$

The number of samples $n$ needed to compute the approximation of the smoothed gradient and the standard deviation $\sigma^2$ are hyperparameters for their method. To transpose this method to the wavelet domain, we add noise to the input before computing its wavelet transform. We refer to this method as $\mathrm{WAM}_{SG}$ throughout the rest of the paper. In appendix A.1, we illustrate the enhancement of the quality of the explanation after applying the smoothing to the gradients as described in equation 2.

**Path integration.** Another approach to derive smooth explanations from the model's gradients consists in averaging the gradient values along the path from a baseline state to the current value. The baseline state is often set to zero, representing the complete absence of features. This technique, introduced by Sundararajan et al. (2017), satisfies two axioms, *sensitivity* and *implementation invariance*. Sensitivity states that "*for every input and baseline that differ in one feature but have different predictions, then the differing feature should be given a non-zero attribution*" and Implementation Invariance that "*the attributions are always identical for two functionally equivalent networks*". Following Sundararajan et al. (2017), we adapt the Integrated Gradient method from the image domain to the wavelet domain. Denoting $\boldsymbol{z} = \mathcal{W}(\boldsymbol{x})$, we evaluate

$$\gamma_{\mathrm{Ig}} = (\boldsymbol{z} - \boldsymbol{z}_0) \cdot \int_0^1 \frac{\partial \boldsymbol{f}_c\left(\mathcal{W}^{-1}\left(\boldsymbol{z}_0 + \alpha(\boldsymbol{z} - \boldsymbol{z}_0)\right)\right)}{\partial \boldsymbol{z}} \mathrm{d}\alpha, \tag{3}$$

where $\boldsymbol{z}_0$ denotes the baseline state of the wavelet decomposition of $\boldsymbol{x}$. We refer to this implementation of WAM as $\mathrm{WAM}_{IG}$. In appendix A.1, we illustrate the enhancement of the quality of the explanation after applying the smoothing to the gradients as described in Equation 2 and after integrating the gradients, as described in Equation 3. We also discuss the visualization properties that emerge when using either method.

## 3.2 DECLINATIONS OF THE WAM FOR EACH MODALITIES AND THEIR PROPERTIES

**Practical implementation.** The Wavelet Attribution Method (WAM) is computed across various modalities by leveraging wavelet transforms to analyze model sensitivity. For **images**, the WAM is visualized on the dyadic wavelet transform, allowing decomposition of important coefficients at each scale and reconstruction of regions critical to model predictions. See Figure 2 for a visualization of the WAM in the 2D setting. For **audio**, the WAM computes sensitivity to wavelet coefficients and mel-spectrogram gradients, bridging waveform and spectrogram-based explanation methods. For **3D shapes**, the WAM is applied to voxel grids using 3D wavelet transforms, extending wavelet analysis to structured data. This unified approach provides insights into model behavior

across modalities, enabling meaningful perturbations, minimal image analysis, and connections to frequency-centric robustness frameworks. In section 3.2 and in appendix C, we highlight the connections enabled by our method.

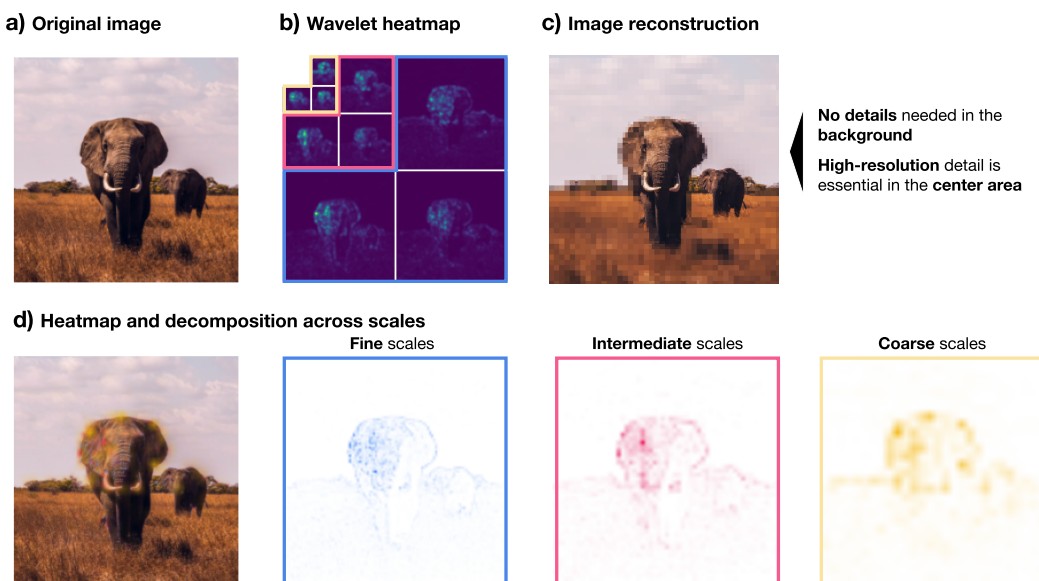

Figure 2: **WAM for images.** Our method decomposes the important components at different scales (i.e., different levels of details) and enables us to see *what* is seen on the image by the model.

By leveraging the wavelet domain, our approach addresses the challenges associated with pixel-space optimization and provides a more comprehensive understanding of the model's behavior. This method generalizes across different data modalities and can be a valuable tool for interpreting complex neural networks in various applications.

**Images: frequency-centric perspectives on model robustness.** Scales in the wavelet domain correspond to dyadic frequency ranges in the Fourier domain. Several works documented a correlation between the reliance on low frequency to make predictions and the robustness of the model (Zhang et al., 2022; Chen et al., 2022; Wang et al., 2020). We can leverage WAM to characterize a model's robustness, thus connecting feature attribution and robustness. On Figure 3, we evaluate the reliance on the different scales by summing the importance of each component within each scale. We average this importance over 1,000 images and thus obtain the average importance of each scale for a model's prediction. We compare a vanilla ResNet-50 (He et al., 2016) with three adversarially robust models : ADV (Madry et al., 2018), ADV-Fast (Wong et al., 2020) and ADV-Free (Shafahi et al., 2019). We can see that adversarially robust models rely more on the coarsest scale (leftmost bars on Figure 3) than the vanilla ResNet-50. On the other hand, they rely less on the finest scales (i.e., the highest frequencies, corresponding to the rightmost bars on Figure 3), thus backing the existing results established in the Fourier domain. Therefore, WAM can be used to characterize the robustness of a model.

**Audio: post-hoc identification of relevant parts of the input audio.** Figure 4 qualitatively illustrates an application of WAM for audio signals. Herein, we perform a noise experiment to add 0 dB white noise to a target audio to form the input audio. The models prediction does not alter after introducing the noise and thus the model is expected to still rely on parts of input audio coming from target audio for its decision. The interpretation audio in Figure 14 generated using top wavelet coefficients provides insights into the decision process and supports this hypothesis. In particular, it almost entirely filters out corruption audio and without requiring any training it also clearly emphasizes key parts of target audio. Similarly, we discuss in appendix C how WAM also retrieves the key parts of an audio signal that has been corrupted with another source (overlap experiment).

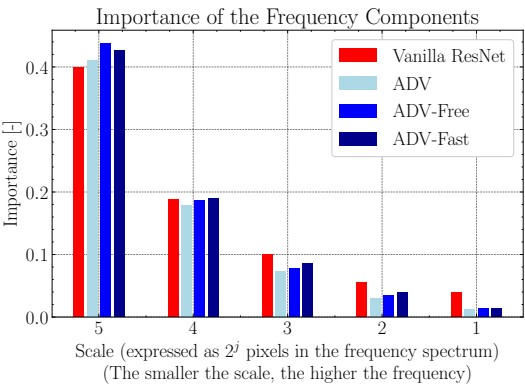

Figure 3: Assessment of model's robustness with the WAM.

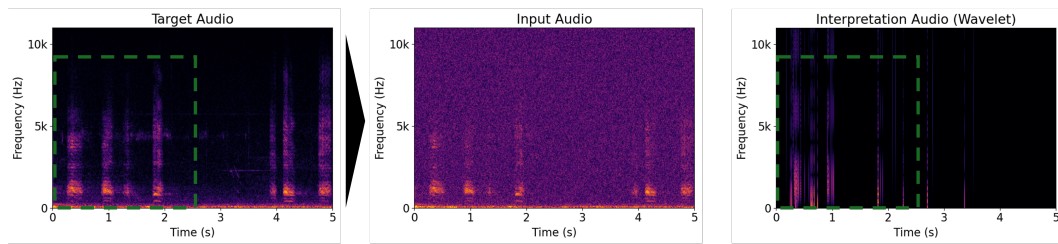

Figure 4: Qualitative illustration of WAM for audio via a Noise experiment. We add 0 dB white noise on he audio of the target class ('Crow') to form the input to the classifier. Interpretation audio reconstructed with important wavelet coefficients virtually eliminate noise, and also clearly emphasize parts of the target class audio (indicated with green box).

**Explanations on voxels.** We retrieve on voxels the same decomposition as for images or audios. Figure 5 highlights the significance of the edges at larger scales, whereas at smaller scales, the importance becomes increasingly concentrated at the center of the digit. To the best of our knowledge, WAM is the first method to show such decomposition on shapes.

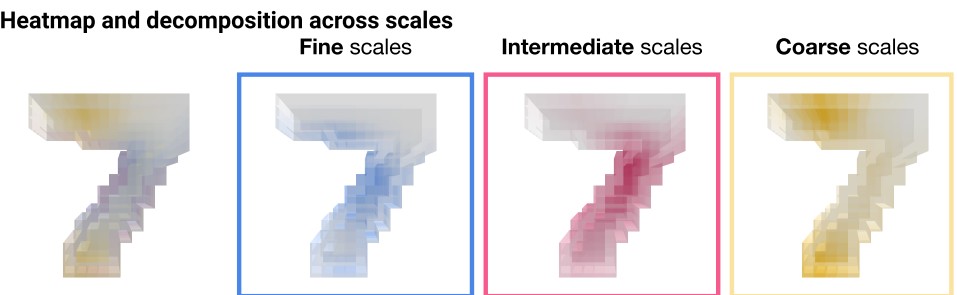

Figure 5: Decomposition of the different important scales on a voxel with the WAM.

# 4 RESULTS

## 4.1 EVALUATION SETTING

We evaluate WAM in two distinct settings: images and audio. Evaluation is carried out on usual benchmarks for both modalities. We do not consider the 3D setting for quantitative evaluation due to the lack of comparable baselines.

**Common evaluation metrics.** We quantitatively assess the accuracy of our method by leveraging the **Faithfulness** (Muzellec et al., 2023), defined as the difference between the **Insertion** and the **Deletion** scores, introduced by Petsiuk et al. (2018). Insertion and Deletion have been widely used in XAI to evaluate the quality of feature attribution methods (Fong & Vedaldi, 2017). The Deletion measures the evolution of the prediction probability when one incrementally removes features by replacing them with a baseline value according to their attribution score. Insertion consists in gradually inserting features into a baseline input. Samek et al. (2016) and Li et al. (2022) have shown that Faithfulness is effective in evaluating attribution methods. Given a model $f$ and an explanation functional $\gamma$, the Faithfulness $F$ is given by

$$F(f, \gamma) = \text{Ins}(f, \gamma) - \text{Del}(f, \gamma). \tag{4}$$

We provide a detailed derivation of the Insertion and the Deletion scores in appendix B.1. Insertion and Deletion were initially defined in the context of images, but we propose a definition that expands them to audio.

**Modality-specific metrics.** In addition to the Faithfulness, we compare WAM using the $\mu$-**Fidelity** (Bhatt et al., 2021) for images and the **Faithfulness on Spectra (FF**, Parekh et al., 2022**)** and **Input Fidelity (Fid-In**, Paissan et al., 2023**)** for audio. We refer the reader to the appendix B.2 for a thorough definition of these metrics and a discussion of the results.

**Evaluation setting for images.** For images, we evaluate our method on a subset of the validation set of ImageNet (Russakovsky et al., 2015). Our subset contains 1,000 images randomly sampled from the 50,000 images of the validation set of ImageNet. We consider four model architectures representative of the popular topologies currently used. We consider the following models: the ResNet (He et al., 2016), the ConvNext (Liu et al., 2022), the EfficientNet (Tan & Le, 2019) and the Data efficient transformer (DeiT, Touvron et al., 2021). We refer the reader to the appendix A.2 for more details on the model's parametrizations that we used. This evaluation framework is based on the frameworks of Fel et al. (2021).

We compare our method with alternative gradient-based methods, namely Saliency (Simonyan et al., 2014), Integrated Gradients (Sundararajan et al., 2017), GradCAM, GradCAM++ and Guided Backpropagation (Selvaraju et al., 2017), and SmoothGrad Smilkov et al. (2017). We focus only on gradient-based methods, as they are more faithful and faster to generate than alternative approaches (Crabbé & van der Schaar, 2023; Wang & Wang, 2021; Xue et al., 2023).

**Evaluation setting for audio.** We evaluate our method on the dataset for Environmental Sound Classification (ESC-50, Piczak, 2015). We pick the 400 samples of the first fold of ESC-50, as our backbone model has been trained on the remainder of the dataset, and evaluate the CNN classification model of Kumar et al. (2018) as our black-box model to explain. We consider a single model as alternative models (Huang & Leanos, 2018; Wilkinghoff, 2021; Lopez-Meyer et al., 2021) are only variations around the same topology. We consider two variants of the ESC-50 dataset: the original (unaltered samples) and noisy samples, for which we add 0 dB white noise to the input samples whose prediction we seek to explain. We include three baseline methods, which return explanations on the mel-spectrogram of the input samples: the GradCAM (Selvaraju et al., 2017), Integrated Gradients (Sundararajan et al., 2017), SmoothGrad (Smilkov et al., 2017) and Saliency (Simonyan et al., 2014).

### 4.2 QUANTITATIVE EVALUATION RESULTS

**Images.** As displayed on Table 1, we can see that WAM for 2D signals outperforms competing baselines according to the Faithfulness metric. In appendix B.2, we present additional results using the Insertion, Deletion, and the $\mu$-Fidelity. The good results are mostly driven by the fact that WAM performs well in terms of Insertion. WAM also passes the randomization test (Adebayo et al., 2018). We refer the reader to appendix B.3 for more details on this test.

**Audio.** Table 2 presents the evaluation results for audio. For WAM, we generate the explanations from the wavelet coefficients. We can see that for audios, WAM also achieves state-of-the-art results

and outperforms the competing metrics in terms of Faithfulness of spectra, Input fidelity and Insertion. The results of the other metrics are in line with those of competing approaches. In appendix B.2, we provide additional results where explanations are computed from the mel-spectrogram of the waveform. In this case, we report that WAM's performance is more in line with competing approaches, thus showing the added value brought by explaining the model's decision through the wavelet domain.

Table 1: **Faithfulness** (Muzellec et al., 2023) score obtained on 1,000 images from the validation set of ImageNet and for different model architectures. Higher is better. **Bolded** results are the best and underlined values are the second best.

| Method | *ResNet* | *ConvNext* | *EfficientNet* | *DeiT* | Mean |
|---|---|---|---|---|---|
| Saliency | 0.025 | 0.032 | 0.008 | 0.038 | 0.025 |
| Integrated Gradients | 0.000 | 0.001 | 0.000 | 0.003 | 0.001 |
| GradCAM | 0.134 | 0.072 | 0.061 | 0.162 | 0.107 |
| GradCAM++ | 0.184 | 0.055 | 0.050 | 0.044 | 0.083 |
| SmoothGrad | 0.023 | 0.000 | 0.010 | 0.004 | 0.009 |
| Guided-Backpropagation | 0.001 | 0.001 | 0.001 | 0.000 | 0.000 |
| $WAM_{SG}$ (ours) | **0.438** | **0.334** | **0.350** | **0.423** | **0.386** |
| $WAM_{IG}$ (ours) | 0.344 | 0.359 | 0.370 | 0.420 | 0.373 |

Table 2: **Evaluation scores** of WAM and comparison with baselines on 400 audio samples from ESC-50 (fold 1). The column "ESC" indicates that the samples are unaltered. The column "+WN" indicates that the samples have 0 dB Gaussian white noise. We report the results with explanations generated from the wavelet coefficients of the waveform. **Bolded** results are the best and underlined values are the second best.

| Method | Faithfulness (↑) | | Insertion (↑) | | Deletion(↓) | | FF (↑) | | Fid-In (↑) | |
|---|---|---|---|---|---|---|---|---|---|---|
| | ESC50 | +WN | ESC50 | +WN | ESC50 | +WN | ESC50 | +WN | ESC50 | +WN |
| IntegratedGradients | **0.264** | **0.310** | 0.267 | 0.312 | **0.047** | **0.045** | **0.207** | **0.207** | 0.220 | 0.225 |
| GradCAM | 0.072 | 0.073 | 0.274 | 0.274 | 0.201 | 0.201 | 0.137 | 0.135 | 0.517 | 0.542 |
| Saliency | 0.066 | 0.065 | 0.220 | 0.221 | 0.154 | 0.156 | 0.166 | 0.168 | 0.253 | 0.245 |
| SmoothGrad | 0.184 | 0.184 | 0.251 | 0.251 | 0.067 | 0.067 | 0.193 | 0.194 | 0.177 | 0.175 |
| $WAM_{SG}$ (ours) | 0.197 | 0.205 | **0.449** | **0.452** | 0.252 | 0.246 | 0.132 | 0.130 | **0.718** | **0.690** |
| $WAM_{IG}$ (ours) | 0.176 | 0.182 | 0.436 | 0.442 | 0.260 | 0.261 | 0.118 | 0.124 | 0.652 | 0.657 |

## 5 DISCUSSION

**Conclusion.** We have introduced a novel approach for feature attribution by computing explanations in the wavelet domain rather than the input domain, providing a framework applicable to audio, images, and shapes. This method shifts away from traditional pixel-based decompositions used in saliency mapping, offering more precise insights into model decisions by leveraging the wavelet domain's ability to preserve inter-scale dependencies. This ensures that critical aspects like frequency and spatial structures are maintained, resulting in richer explanations compared to traditional feature attribution methods.

Our method, WAM, shows a strong ability to highlight essential audio components in noisy samples, isolate necessary shape and texture features for accurate predictions, and offer richer explanations for shape classification. Quantitatively, it achieves state-of-the-art results across both audio and image benchmarks.

**Limitations & future works.** Despite its advantages, the current method does not extend to 3D point cloud data, and for audio, the greedy extraction of important coefficients is unsuitable for generating listenable explanations. Future work could explore alternative wavelet decompositions, such

as continuous or complex wavelets for audio explanations and graph wavelet transforms to handle unstructured point clouds. Additionally, our method could be applied to videos mathematically similar to the voxel data used in this work. We hope this approach will inspire further research into the properties of explanation domains, the wavelet domain being one such domain.

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
