# OpenReview forum: "One Wave to Explain Them All: A Unifying Perspective on Post-hoc Explainability"
_ICLR.cc/2025/Conference — Submitted to ICLR 2025_

### Official Review · Reviewer_AcnY · 2024-10-21

**Soundness:** 1
**Presentation:** 2
**Contribution:** 2
**Rating:** 5
**Confidence:** 3

**Summary:**

In this paper, the authors identified a gap in existing attribution methods, specifically their inability to explain the structural components of input data. The authors propose a novel wavelet-based attribution method extending to multiple modalities including images, audio, and 3D shapes. While the topic is timely and the problem addressed is of significant importance, the proposed method lacks a clearly demonstrated advantage in explaining structural components compared to existing techniques. Moreover, the quantitative results do not convincingly showcase the method's superiority. Therefore, I tend to reject this paper in its current version.

**Strengths:**

1. The topic addressed in this paper is important, and the proposed method is novel. Current attribution methods struggle to explain and distinguish structural components in the input, and the integration of wavelets into attribution calculations shows promise in addressing this limitation.
2. Extensive evaluations are conducted across multiple modalities, including images, audio, and 3D shapes.
3. This paper is easy to follow, with detailed descriptions of evaluation metrics and experiments.

**Weaknesses:**

1. The primary weakness is the lack of significant differentiation between the proposed attribution method and existing approaches. The paper provides limited analysis or visualizations that convincingly show how WAM offers better hierarchical explanations. While some comparisons are presented (e.g., in Figures 2 and 12), I don’t see clear and enough advantages in explaining input structures. The authors should further explore or emphasize the distinctive aspects of their method.
2. The quantitative results do not consistently demonstrate improvements over existing attribution methods. While comparisons with older methods are acceptable given the novelty of the proposed approach, the proposed method falls significantly behind in several key metrics, such as in Tables 3 and 5 (Appendix). Additionally, the results in Table 1 are concerning; given the definition of the Faithfulness metric in Eq. 9, the output should always be positive. Why, then, are many results in Table 3 reported as zero?
3. The organization of the results section could be improved. For example, the discussion of perturbation-based attribution methods in Section 4.2 appears abruptly and feels disconnected from Section 4.1.

**Questions:**

1. Could you clarify why many results in Table 3 are zero when using the Faithfulness metric?
2. Could you explain why Integrated Gradients perform best on the \mu-Fidelity metric but perform worst in Faithfulness? Is this discrepancy due to different experimental setups?

---

### Official Review · Reviewer_kXsu · 2024-10-27

**Soundness:** 2
**Presentation:** 3
**Contribution:** 2
**Rating:** 5
**Confidence:** 3

**Summary:**

This paper introduces the “Wavelet Attribution Method(WAM),” a feature attribution technique in explanable AI that attiburion is on wavelet domain. WAM leverages the wavelet domain to extend gradient-based feature attributions, preserving the multi-scale structures of the input data. This approach provides a unified framework applicable across various modalities such as images, audio, and 3D shapes. Empirical evaluations demonstrate that WAM matches or surpasses other methods in faithfulness metrics.

**Strengths:**

The paper is well-structured, presenting its concepts clearly and in an accessible manner. The theoretical foundations for using gradients in the wavelet domain are well-developed, filling a gap in the current literature where such an approach has not been extensively explored.

 The paper introduces a novel method by leveraging gradients in the wavelet domain, which provides a new perspective on feature attribution.

The paper's evaluation of the proposed method's faithfulness using multiple faithfulness metrics is thorough and valuable. By comparing the proposed approach across different evaluation criteria, the authors demonstrate the robustness and reliability of their method.

 The figures and visualizations are well-designed, enhancing the clarity of the paper. They effectively illustrate the principles of the Wavelet Attribution Method (WAM) and provide a clear understanding of how wavelet-based attributions differ from traditional pixel-based methods. The use of visual examples makes the theoretical concepts more accessible and supports the argument for the method’s efficacy.

**Weaknesses:**

The approach used in this paper shares similarities with the WaveletX(Kolek et al, 2022) method, which also performs saliency mapping in the wavelet domain. The primary distinction lies in the use of gradients as a mask in this work, While WaveletX optimize the mask on wavelet domain. However, this difference may not be significant enough to constitute a radical contribution to the field. It would be better to explicitly compare both methods and highlight the novel aspects of WAM.

The paper does not include quantitative assessments for 3D shape analysis and relies solely on qualitative results. Incorporating quantitative metrics would strengthen the evaluation and provide a more comprehensive understanding of the method's performance in this domain.

Although the authors claim that the Wavelet Attribution Method (WAM) outperforms other approaches across different domains, the results in Table 2 suggest otherwise. Specifically, WAM does not consistently outperform Integrated Gradients, indicating that the performance advantage may not be as significant as claimed.

The experimental comparisons primarily involve methods like Integrated Gradients, GradCAM++, and SmoothGrad, which are not the most recent or best-performing approaches according to the fidelity metric. Including comparisons with more recent and state-of-the-art methods, such as LRP-αβ (Samek et al., 2016), LayerCAM (Jiang et al., 2021), Guided Backpropagation (Selvaraju et al., 2016), AttnLRP (Achibat et al., 2024), and SRD (Han et al., 2024), would strengthen the evaluation and better demonstrate WAM's superiority.

The paper does not adequately demonstrate how WAM enables an understanding of what features the model uses to make decisions. While the method highlights important wavelets, it does not clarify the specific meaning or relevance of these wavelets to the classification task. For instance, approaches like CRAFT (Concept Recursive Activation FacTorization for Explainability, Fel et al., 2023) offer more explicit explanations by identifying meaningful concepts (e.g., "elephant tusk") that recur across multiple samples. Providing a similar level of interpretability by linking important wavelets to specific semantic features would improve the explanatory power of WAM.

**Questions:**

In Figure 2 (d), I don’t know why decomposing important coefficient at each scale is necessary. Is there specific reason that scaling is important?

Does WAM can discriminate each object when multiple classes are at one image, such as Cat-dog image?

What is GRADWCAM in figure 5?

**Details Of Ethics Concerns:**

Potential plagiarism: this paper's approach is almost identical with the paper "Assessment of the Reliablity of a Model's Decision by Generalizing Attribution to the Wavelet Domain" which presented at workshop "XAI in Action: Past, Present, and Future Applications workshop at NeurIPS 2023". I understand that workshop papers are not considered as formal archived publications. However, I have some concerns about potential plagiarism, as it is unclear whether the authors of the current submission are the same as those of the workshop paper.

---

### Official Review · Reviewer_AFY7 · 2024-10-30

**Soundness:** 3
**Presentation:** 3
**Contribution:** 2
**Rating:** 5
**Confidence:** 5

**Summary:**

The paper presents the Wavelet Attribution Method (WAM), a novel explainability approach for deep learning models in the wavelet domain. Unlike traditional pixel-based attribution methods (saliency maps), WAM leverages the structural benefits of the wavelet domain, offering a more generalizable approach that applies to various data modalities, including images, audio, and 3D shapes. WAM decomposes the model’s decisions by analyzing how features in the wavelet-transformed space affect predictions. The method integrates gradient-based attribution techniques with wavelet decomposition to capture both where (location) and what (content) aspects of the data structure. Through empirical evaluation, WAM demonstrates superior performance on faithfulness and fidelity metrics for image and audio data, achieving enhanced clarity in the model’s decision-making process.

**Strengths:**

WAM brings an intriguing approach by utilizing wavelet decomposition to improve gradient-based feature attribution methods. The method leverages the mathematical properties of the wavelet domain, potentially addressing limitations of saliency maps that flatten hierarchical and spatial relationships. This could provide meaningful explanations by capturing features across multiple scales. In theory, WAM’s emphasis on inter-scale dependencies could enhance explainability across images, audio, and 3D shapes, offering an innovative view on XAI. Additionally, by unifying SmoothGrad and Integrated Gradients, WAM capitalizes on established approaches while potentially broadening their applicability across multiple modalities. This multimodal capability, though perhaps overstated, should be a promising generalization that is not commonly found in the comparison methods, which are often restricted to single data domains.

**Weaknesses:**

Despite its ambitious goals, WAM introduces several ambiguities and potential oversights. Key among them is the unclear use of 'structural components', a term the paper uses to describe feature-level insights that the method claims to provide. This concept, critical to WAM’s claims of 'what' explainability, lacks a clear definition or grounding in quantifiable relationships among components, making it difficult to ascertain whether these are indeed structural features rather than just relevant input attributes. Furthermore, while wavelet decomposition is introduced as a novel approach to attribution, the practical interpretability of multi-scale heatmaps remains underexplored in the paper. it is unclear how users can derive specific insights from these maps without a more explicit explanation. WAM’s assertion of state-of-the-art (SOTA) performance is another potential weakness, given that its comparisons rely largely on 2017 models like SmoothGrad, Grad-CAM, and Integrated Gradients, raising questions about whether the method is genuinely competitive in the context of more recent advancements in the XAI field. Additionally, the effectiveness of the faithfulness metrics used to benchmark WAM’s performance could benefit from further clarification, especially given the method’s claims of surpassing existing techniques across domains.

**Questions:**

1. Could you provide a more rigorous definition of 'structural components' and clarify how they differ from standard features in the context of explainability? Specifically, can we establish any meaningful relationships between these components based on their implicit structure?

2. Since inter-scale dependencies are central to your claims, what specific dependencies does the wavelet domain preserve, and how does this preservation impact attribution in practice? For example, in the case of image explanations, presenting different scales of explanations does not seem to provide substantial additional insight.

3. To what extent have you included newer, state-of-the-art models in your evaluation, and how might WAM perform with models developed after 2017, considering the rapid advancements in explainability techniques? Have you considered expanding the method to a self-explainable framework by introducing a novel loss term directly in the wavelet domain?

---

### Official Review · Reviewer_F8CS · 2024-11-03

**Soundness:** 3
**Presentation:** 3
**Contribution:** 2
**Rating:** 5
**Confidence:** 3

**Summary:**

This paper present the Wavelet Attribution Method (WAM), which improves gradient-based feature attribution by utilizing the wavelet domain as a comprehensive framework for explaining classifiers across various domains. The findings indicate that WAM meets or surpasses SOTA metrics for faithfulness, effectively identifying not only the locations of significant features but also highlighting their structural patterns.

**Strengths:**

1. Developing a multi-domain explanation method is intriguing, and the discussion of key challenges is reasonable.
2. The manuscript is well written. It is easy to follow this work.
3. The experiments conducted with the proposed methods are adequate.

**Weaknesses:**

1. I believe there is a lack of sufficient baselines. It would be helpful to include more options such as LIME, SHAP, and concept-based explanations for image and audio data. Since there is no quantitative evaluation in 3D settings, adding 3D LIME, SHAP, sensitivity analysis, and Layer-wise Relevance Propagation (LRP) for 3D baselines would be a solid starting point.

   References:

   [1] "Why Should I Trust You?": Explaining the Predictions of Any Classifier
   [2] A Unified Approach to Interpreting Model Predictions
   [3] Towards Automatic Concept-based Explanations
2. The experiments were conducted on only one dataset; therefore, it would be essential to include results from several datasets.
3. In the audio results (Figure 1 and Figure 4), it is quite challenging to identify the areas being explained. Making the less important areas grayscale while highlighting the significant areas in red would improve interpretability.
4. It would have been better to conduct a human study for the qualitative evaluation. For example, utilizing Amazon Mechanical Turk (MTurk) to ask annotators to evaluate WAM while providing explanations for other baselines would be beneficial.

**Questions:**

Please see Weaknesses.

---

### Meta-Review · Area_Chair_J95M · 2024-12-16

**Metareview:**

This work uses wavelet decompositions and gradient-based attribution techniques to capture location and content of the data used by a classifier to make a decision.

After reading reviews and the author's rebuttal, I see there are some remaining concerns to be addressed:

* AFY7: Concerned about the amount of contribution/impact of this work.
* kXsu: Remains concerned about the practicality of this work from a user standpoint.
* AcnY: Remains concerned about the quality of the experimental evaluation.
* F8CS: Decided to keep their score after the author rebuttal without justification.

**Additional Comments On Reviewer Discussion:**

During the reviewer / AC discussion AFY and AcnY reaffirmed their concerns and thus I cannot support the acceptance of this work.

---

### Decision · Program_Chairs · 2025-01-22

Reject